# Are Veterinary Students Using Technologies and Online Learning Resources for Didactic Training? A Mini-Meta Analysis

Edlira Muca [1,*], Damiano Cavallini [2], Rosangela Odore [1], Mario Baratta [3], Domenico Bergero [1] and Emanuela Valle [1]

1   Department of Veterinary Sciences, University of Turin, 10095 Grugliasco, Italy
2   Department of Veterinary Science, University of Bologna, 40064 Ozzano dell'Emilia, Italy
3   Department of Chemistry, Life Sciences and Environmental Sustainability, University of Parma, 43124 Parma, Italy
*   Correspondence: edlira.muca@unito.it

**Abstract:** Over the last years, there has been an increase in online educational resources and media device use for educational purposes in veterinary settings. However, an overall analysis of these studies providing measurements of the use of learning resources and media devices could be particularly useful for veterinary teachers. The evolution of technology, coupled with the advent of pandemic-related restrictions in person lessons, has made it imperative that educators consider how students may access educational material, as well as what type of educational material may be available to them. Databases including PubMed, Scopus, CAB Abstracts, and Web of Sciences were searched for relevant studies from January 2012 to June 2022. A mini-meta-analysis for proportions was performed using RStudio. Results highlight a high use of portable media devices with differences among countries, continued good use of traditional textbooks, moderate use of online tools, and low use of research papers. The results suggest that despite living in a technologically advanced world, veterinary students have attitudes towards digital resources that cannot be assumed.

**Keywords:** technology; didactic training; online learning; electronic devices; veterinary students





## 1. Introduction

Over the past decades, there has been an increase in online educational resources available for higher education [1,2] as well for veterinary medical education [3,4]. Along with traditional learning sources such as textbooks and face-to-face lectures, veterinary students are increasingly using electronic devices and a plethora of online material for integrating their learning [3,5]. Recently, with advancements in technology, several educational websites, massive open online courses (MOOCs), online professional networks, virtual educational conferences, and online academies have been established [6–14]. Contemporarily, the introduction of cutting-edge technologies, including smartphones, laptops, tablets, computers, wearable devices, and faster internet access, has also revolutionized the field of veterinary science and veterinary education [15,16]. This has resulted in an expansion of electronic device usage in educational settings altogether, with an augmentation of online resources accessed by veterinary students [3,17,18]. The global spread of COVID-19 has accelerated the integration of digital devices and online resources in veterinary medical education [19–22]. Moreover, the advance of science and technology has increased the application of digital technologies (DT) and artificial intelligence tools in medical and veterinary medicine teaching [23]. Furthermore, today's innovative teaching and learning methods seem to rely mainly on the integration of information and communication technologies (ICT) [24–27]. Therefore, the importance of online resources and digital devices are likely to persist in the post-COVID era. A further boost to the use of digital technologies in

veterinary education comes from the European Association of Establishments for Veterinary Education (EAEVE). The experts of the European Coordinating Committee on Veterinary Training (ECCVT) recently underlined that the new generation of students is familiar with digital interactions, and universities need to ensure the use of such technologies in educational programs [28].

The evolution of technology, coupled with the advent of pandemic-related restrictions on in-person lessons, has made it imperative that educators consider how students may access educational material, as well as what type of educational material may be available to them. Therefore, the aim of the current study was to conduct a systematic review and a mini meta-analysis of findings concerning the use of electronic devices and online and/or traditional resources for educational purposes among global veterinary students.

## 2. Materials and Methods

### 2.1. Systematic Review

In order to summarize the existing pieces of evidence pertaining to this subject, a systematic review of empirical studies of multiple designs was performed.

### 2.2. Eligibility Criteria

We included any study that met the following criteria:

a.  All research articles that reported the use of online resources and electronic devices by veterinary students for study purposes only.
b.  Cross-sectional studies that assessed as a primary or secondary outcome the use of electronic devices to access the learning environment and the usage of learning resources in any format.
c.  Respondents were veterinary students from undergraduate to residency level.
d.  Studies that included surveys or research-based projects.
e.  Written in English language only.
f.  Published from 1 January 2012 to 10 June 2022.
g.  Peer-reviewed only. The exclusion criteria were studies in which the use of online resources and electronic devices were not used for educational purposes. Commentaries, letters to editor, editorials, expert opinions, original articles without sufficient details, reviews, and conference abstracts or proceedings were also excluded.

### 2.3. Data Sources and Search Strategy

To obtain the best coverage of the veterinary literature [29], we systematically searched the following databases, CAB Abstracts, Scopus, MEDLINE (via PubMed), and Web of Science. In addition, a full reference list of the included articles was checked to identify additional relevant studies. The meta-analysis included studies published over a 10-year period (1 January 2012 to 10 June 2022). A combination of keywords related to "online learning", "online resources" "blended learning", "hybrid learning", "distance learning", "electronic devices", "veterinary education", and "veterinary students" was used. Initially, a search strategy was developed for the PubMed database, and this strategy was then adapted for all other databases for their inception (See Table 1 for complete search strategy for PubMed). The truncation symbol (*) was applied to ensure that all the words related were included in the search.

### 2.4. Study Selection

Duplicates of all articles retrieved were removed and screened for full-text review if they were original research studies that assessed the use of learning resources and type of electronic devices to access learning environments and/or online educational resources or materials.

Two reviewers (E.M. and D.C.) independently screened titles and abstracts of all retrieved articles and then completed a full-text article screening based on inclusion and

exclusion criteria. Disagreements were solved by discussion and a third reviewer (E.V.), who further decided if the article should or not be included.

**Table 1.** A summary of the search strategy adopted in the present study.

| Search Strategy Item | Search Strategy Details |
|---|---|
| String of Keywords | (education, veterinary [mh] OR veterinary, students [mh] OR veterinary, education [mh] OR "undergraduate veterinary education" OR "veterinary students" OR "veterinary student" OR "veterinary schools" OR " veterinary school") AND ("online learn*" OR "electronic learn*" OR "e-learn*" OR "distance learn*" OR "flipped learn*" OR "hybrid learn*" OR "blended learn*" OR "mobile learn*" OR "m-learn*" OR "digital learn*" OR "online resourc*" OR "smartphone" OR "laptop" OR "tablet" OR "desktop" OR "computer" OR "online participation" OR "online discussion" OR "electronic devic*" OR "digital devic*" OR "educational web*" OR "social media" OR "video" OR "multimedia" AND (measure* OR assess* OR evaluate*) |
| Searched Databases | PubMed, Web of Science, CAB Abstracts, Scopus. |
| Time Filter | From 1 January 2012 to 10 June 2022 |
| Language Filter | English |

*2.5. Data Extraction*

The same reviewers (E.M. and D.C.) independently performed data extraction on each study using the same inclusion and exclusion criteria. Disagreements were solved by discussion. The information that was extracted included the name of the authors, year of publication, study design, number of participants, and types of electronic devices and learning resources used only for learning purposes. Moreover, data regarding types of educational resources (YouTube videos, social media platforms, educational applications, research papers, e-books and textbooks) were extracted. Microsoft Excel was used for data management.

*2.6. Terminology*

- Non-portable electronic devices (e.g., desktop computers) can be defined as any type of media device designed for regular use at a single location [30].
- Portable electronic devices are defined as any media device type with capacities to store, record, transmit text/videos/audios. Examples of such devices are smartphones, laptops, tablets, etc. These devices offer features of portability, which desktop computers cannot offer [31].
- Textbooks are defined as books used as a standard work for studying a particular subject [32].
- E-Books or electronic books are electronic versions of printed books that can be read on computers or handheld devices which are designed specifically for them [33].
- Educational websites are defined as appropriately designed and developed websites which hold the potential to provide to students valuable educational content [34].
- Educational applications are defined as educational software which are specifically designed and developed for teaching and learning purposes [35].
- Research papers are defined as manuscripts that represent original works of scientific research or studies [36].
- YouTube videos are defined as visual content shared through a channel called YouTube. Due to its open-access nature, content can reach a broad audience, and it is often used in education as a platform for sharing educational videos [18].
- Social media platforms are defined as any sites which combine internet- based technologies and mobile applications and allow users to share content and/or participate in social networking [37].

\* Data extraction was performed in all above-mentioned resources if they were reported by at least three studies and subsequently included in the proportional meta-analysis.

### 2.7. Statistical Analysis

The meta-analysis was conducted via forest plot, which graphically represented the consistency and reliability of the results from selected studies; random and fixed effects meta-analysis models were carried out using the total sample size and number of participants with positive response. In this study, the forest plot was designed with RStudio (v1.3.959) [38] packages "tidyverse", "meta", and "metaphor" [39]. The effect size of each study was computed as an outcome, and pooled effect size was also calculated to observe the heterogeneity among studies. Between-study variations were assessed using (1) the Cochran's Q (chi-squared) test of heterogeneity, to evaluate whether the variation between studies exceeded that expected by chance, where $p \leq 0.1$ indicated significant heterogeneity, and (2) the Higgins $I^2$ statistic, to estimate the percentage of total variation in effect estimates across the studies attributable to heterogeneity rather than chance, where $I^2 > 50\%$ may indicate substantial heterogeneity [40].

## 3. Results

### 3.1. Study Selection

A total of 675 records were identified through initial searching. After removing duplicates, the remaining 359 records were examined by reading the title and the abstract. Of these, 331 records were found to be irrelevant and did not fulfill the eligibility criteria, whereas 28 papers remained for full-text review. Twenty of them were further excluded for the unmet criteria set, leaving eight articles for final qualitative analysis. Finally, six studies that assessed similar type of electronic devices and online resources were eligible for the meta-analysis. Further details are shown in the Flowchart (Figure 1).

### 3.2. Study Characteristics

All eight studies included in the systematic review were cross-sectional surveys conducted in the population of veterinary students, published between July 2017 and January 2022.

Three studies were conducted in United Kingdom [20,41,42]. One study was conducted in Germany [18]. One study was conducted in Egypt [22]. Three surveys were multicenter studies, including participants all over the world, and were carried out in 92 and 87 countries [3,19,21].

Six surveys were conducted online through sharing the questionnaire link via social media platforms and by e-mails, while only one survey was conducted in both modalities with online and paper-based questionnaires. Three studies assessed how COVID-19 affected student learning and performance and the electronic devices that students use to access the learning environment and online educational materials [19,21,22]. Moreover, these studies explored the use of traditional learning materials such as textbooks and/or research papers. Another study was conducted during the COVID-19 lockdown, aiming to assess and evaluate student digital-learning capabilities and the most common behaviors of students during their independent learning time [20]. In addition, this study assessed the most common electronic device used to access the virtual learning environment and for different learning purposes.

Two studies were conducted in specific subjects such as physiology and cardiology among UK veterinary students [41,42]. These studies aimed to evaluate to what extent second- and third-year veterinary students use online learning resources and what electronic devices they use to access these resources. In addition, these studies also assessed the students' preferences toward traditional learning resources such as lectures and recommended textbooks. One survey was conducted among veterinary students all over the world and aimed to assess the use of online learning resources, social media tools, and veterinary-specific educational e-platforms and websites [3]. The study mostly investi-

gated the frequency of use of the above-mentioned resources. One mixed-method study carried out in Germany assessed how veterinary students access online learning materials, especially instructional videos, before clinical skill lab activities [18].

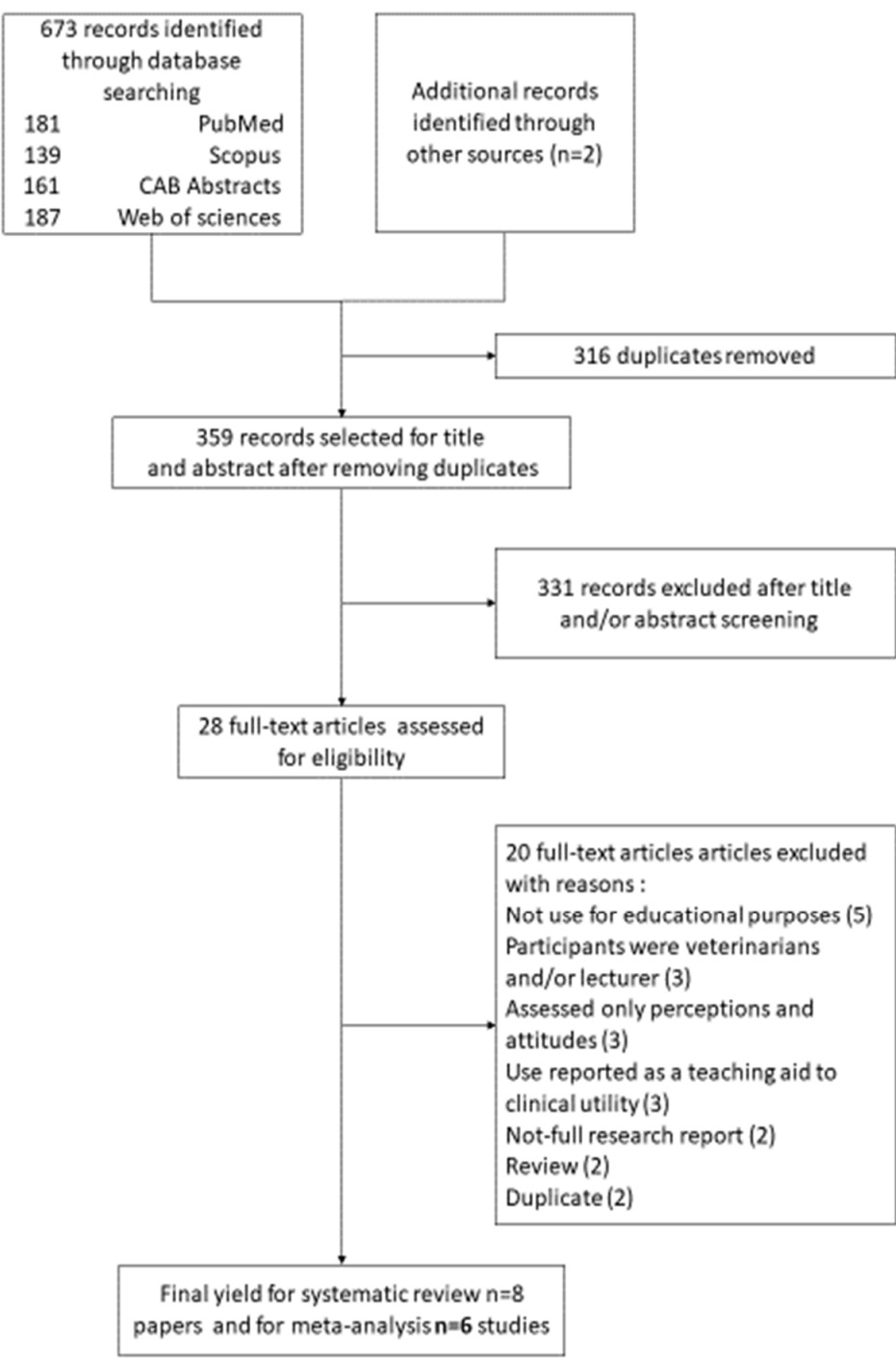

**Figure 1.** Flowchart of study selection proces.

Mean age of respondents varied from (M = 19.57 ± SD = 0.60) to (M = 24.10 ± SD = 5.93).

The origin of participants was as follows: Europe (28.37%), Africa (23.17%), Asia (17.74%), UK (16.5%), North America (7.44%), Oceania (4.84%), South America (1.94%).

A detailed overview of the characteristics of veterinary students involved in the selected studies is shown in Table 2.

Table 3 summarizes the main characteristics of the included studies in systematic review and mini-meta-analysis.

**Table 2.** Main characteristics of respondents involved in the selected studies.

| Study | Nr of Veterinary Students | Gender | | Mean Age | UK | Europe | North America | Oceania | Asia | Africa | South America |
| | | M | F | | | | | | | | |
|---|---|---|---|---|---|---|---|---|---|---|---|
| Gledhill et al. (2017) [3] | 1070 | NS | NS | NS | 326 | 259 | 223 | 119 | 95 | 29 | 14 |
| Müller et al. (2019) [18] | 805 | NS | NS | NS | | 805 | | | | | |
| Mahdy (2020) [19] | 1392 | 674 | 718 | 24.10 ± 5.93 | 32 | 258 | 71 | 56 | 498 | 446 | 31 |
| Sadeeh et al. (2021) [41] | 122 | 12 | 110 | 24–26 | 122 | | | | | | |
| Limniou et al. (2021) [20] | 170 | 33 | 137 | NS | 170 | | | | | | |
| Mahdy & Ewaida (2022) [21] | 961 | 424 | 537 | 22.00 ± 3.42 | 30 | 162 | 96 | 78 | 335 | 234 | 56 |
| Mahdy & Sayed (2022) [22] | 502 | 184 | 318 | 19.07 ± 0.56 | | | | | | 502 | |
| Sadeeh et al. (2022) [42] | 213 | 29 | 184 | 21–23 | 213 | | | | | | |

NS = Not Stated.

**Table 3.** Characteristics of studies included in the systematic review and mini meta-analysis.

| Authors | Study Design | Participants | Type of Devices and Resources Assessed | Measures | Main Findings |
|---|---|---|---|---|---|
| Gledhill et al. (2017) [3] | Cross-sectional (survey-based) | 1070 veterinary students | Ownership of smartphones, tablets, e –readers, laptops and desktop computers. Frequency of use of the following online resources: search engines, MOOCs, virtual worlds, open educational resources (OERs), social networking (e.g., Facebook), social videos (e.g., YouTube), instant messaging (e.g., Messenger), voice calls (e.g., Skype), video conferencing (e.g., Google Hangout), social images (e.g., Pinterest) microblogging (e.g., Twitter), social bookmarking (e.g., Del.ico.us), WikiVet, Merck, VIN, Vetstream, NOVICE. | Questionnaire | The majority of students reported using online educational veterinary resources. Ownership of smartphones was widespread (92%), and the majority of respondents (74%) indicated that the use of mobile devices was essential for their learning. Social media platforms were indicated as essential for collaborating with peers and sharing knowledge between them. The students from less-developed countries were disadvantaged by limited access to technology and networks. |
| Müller et al. (2019) [18] | Mixed-methods (survey-observational) | 835 veterinary students | Ownership of smartphones, laptops, tablets, and computers. Instructional YouTube videos prepared for clinical skill laboratories. | Questionnaire (paper-based and online survey) | Before hands-on activities in the clinical skill laboratories, students watched videos on laptops, tablets, or smartphones. Almost all students rated the instructional videos as valuable and helpful learning tools. |
| vMahdy (2020) [19] | Cross-sectional | 1392 veterinary students | Smartphones, laptops, tablets, and computers. Online classes, PDF lectures, textbooks, YouTube videos, University platforms, educational websites, educational applications, Zoom, WhatsApp groups, Google classroom, social networks, Microsoft teams, Edmodo, Skype, Google Meet, Blackboard, Web Whiteboard, Moodle, WebEx, Canvas, VIN, Edpuzzle, Edverum. | Online questionnaire | 96.7% of students reported that COVID-19 affected their academic performance. However, online instruction provided students with the opportunity for self-directed study. The most challenging aspect of online instruction was related to the hands-on sessions. Zoom was the most-used online tool, followed by WhatsApp groups and Google Classrooms. Online courses were the most preferred sources of online learning. |

**Table 3.** *Cont.*

| Authors | Study Design | Participants | Type of Devices and Resources Assessed | Measures | Main Findings |
|---|---|---|---|---|---|
| Saadeh et al. (2021) [41] | Cross-sectional | 122 veterinary students | Smartphones, laptops, tablets, and computers. Sources for physiology information: lectures, textbooks, random internet search engines, WikiVet, YouTube videos, VIN, Wikipedia, social media platforms and research papers. | Online questionnaire | Traditional resources such as lectures and textbooks were the most-preferred. 97% of students used search engines to supplement their physiology learning. 91.1% of students considered videos to be a valuable tool for their learning. 92% of students indicated that they would first search online for an answer before asking instructors. |
| Limniou et al. (2021) [20] | Cross-sectional | 170 veterinary students | Smartphones and laptops. Word software, presentation software, e-mail packages, statistics packages, spreadsheet software, virtual learning environments, web conferencing applications, video-sharing applications | Online questionnaire | Students reported their most common learning behaviors during the lockdown. Students with high levels of self-regulation and digital literacy reported that they were focused and engaged in their studies during COVID-19 lockdown. |
| Mahdy & Ewaida (2022) [21] | Cross-sectional | 961 veterinary students | Smartphones, laptops, tablets, and computers. YouTube videos, anatomy textbooks, anatomy e-books, educational websites, anatomy Facebook pages, educational applications, anatomy WhatsApp groups, anatomy Telegram channels and research papers. | Online questionnaire | 86% of respondents indicated that they were interested in studying anatomy online during the COVID-19 pandemic. 61% of students were able to understand online anatomy well using online learning resources accessed via electronic devices during the lockdown. |
| Mahdy & Sayed (2022) [22] | Cross-sectional | 502 veterinary students | Smartphones, laptops, tablets, and computers. Anatomy e-books, YouTube videos, Telegram channels, educational websites, Facebook pages, research papers, educational applications and WhatsApp groups. | Online questionnaire | The majority of students were enthusiastic about studying anatomy online during COVID-19 lockdowns. 63% of the respondents were satisfied with the provided learning materials. 66% of the students could understand anatomy through online learning and 67% reported to be comfortable with technological skills. 47% of the respondents believed that online learning of anatomy could replace face-to-face learning. |
| Saadeh et al. (2022) [42] | Cross-sectional | 213 veterinary students | Smartphones, laptops, tablets, and computers. Sources for cardiology information: lectures, textbooks, random internet search engines, WikiVet, YouTube videos, VIN, Wikipedia, social media platforms, and research papers. | Online questionnaire | The lecturer was indicated as the preferred resource and students aged 27 and above preferred recommended textbooks. However, 95.3% of students used search engines for cardiology information and 71.8% of students accessed videos at least once a week for cardiology learning. 93.4% of students indicated that they would search for answers online first rather than contacting the instructor. |

### 3.3. Study Results and Meta-Analyses

Two studies could not be included in the quantitative synthesis and were included in the systematic review only. Both were conducted to assess the use of online learning resources and electronic devices among veterinary students.

One of these studies reported that the usage of online resources and electronic devices was an integrative part of the learning process of veterinary students [3]. Authors found that the ownership of smartphones was widespread and most of the students agreed that the use of electronic devices was essential for their learning. An interesting finding of this study was that students from less-developed countries had limited access to online learning resources [3]. The second study assessed how German students used videos as online learning resources before taking part in clinical skill lab activities and found that online videos were a useful resource to learn clinical skills while contributing to animal welfare by reducing the number of animals used for training purposes [18].

Gledih et al., 2017 [3] assessed the frequency with which students use online learning resources; the assessment of electronic devices (desktop, laptop, tablet, e-reader and smartphone) was based on student ownership rather than use of the above-mentioned devices for learning purposes only. Thus, we decided to not include it.

Finally, the study by Müller et al., 2019 [18] was not included in the meta-analysis because it assessed device ownership among students involved in clinical skill laboratories.

The remaining six studies, which assessed the use of electronic devices and learning resources for study purposes only, were included in the quantitative synthesis.

A combined prevalence of the usage was found for each learning resource chosen. Resources which were reported by at least three studies were included in the proportional meta-analysis. More specifically, the use of non-portable media devices was assessed by a total of five studies [19,21,22,41,42], while the use of portable media devices was evaluated by all the included studies of the mini-meta-analysis [19–22,41,42]. Figures 2 and 3 show the forest plot of the proportion of veterinary students who have reported to use non-portable and portable media devices for study purposes only.

The students who reported use of non-portable media devices to access learning materials and learning environments yielded an average value of 0.099 with a confidence interval of 0.088–0.110 and 0.169 [0.069–0.302] in the fixed and random effects meta-analysis model. The difference is mainly due to the different sample sizes between selected studies. The partial proportions meta-analysis found that compared to the average, Saadeh et al., 2021 and Saadeh et al., 2022 reported greater use of non-portable media devices. In contrast, in Mahdy 2020, Mahdy & Ewaida 2022, and Mahdy & Sayed 2022, when compared to the average lower use of the above-mentioned devices, the students who reported use of portable media devices to access both learning materials and learning environments was high, as expected. The overall average result of the fixed effects model was 0.884 [0.872–0.895], and the random model effects resulted in an average of 0.834 [0.718–0.925].

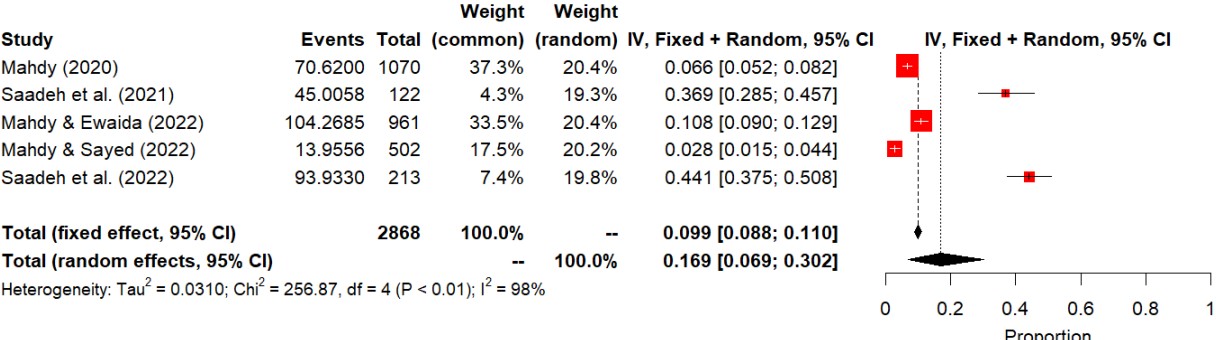

**Figure 2.** The proportion of veterinary students who use non-portable electronic devices for learning purposes. Note: CI = confidence interval 95%; $Tau^2$ = index of effect size dispersion; $Chi^2$ = Cochran's Q (chi-squared) test of heterogeneity; df = degree of freedom; P = *p*-value; $I^2$ = Higgins statistic.

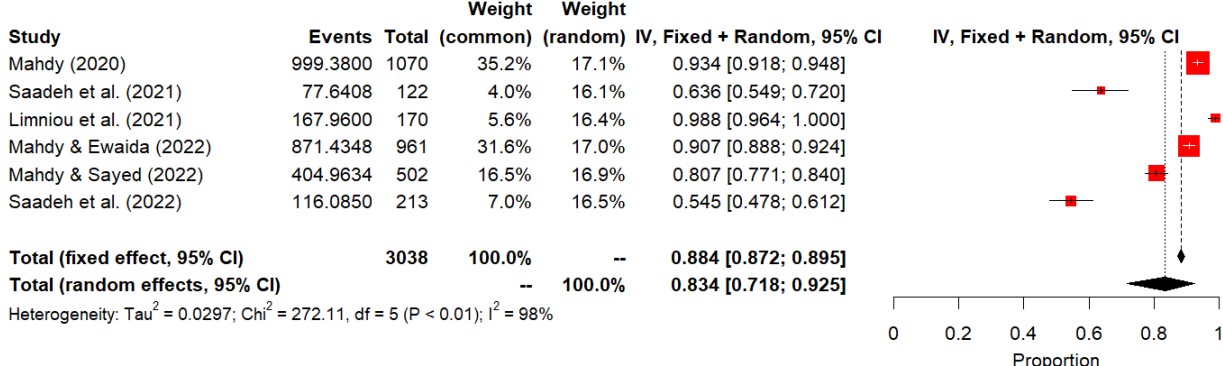

**Figure 3.** The proportion of veterinary students who use portable electronic devices for learning purposes. Note: CI = confidence interval 95%; Tau$^2$ = index of effect size dispersion; Chi$^2$ = Cochran's Q (chi-squared) test of heterogeneity; df = degree of freedom; P = *p*-value; I$^2$ = Higgins statistic.

Overall, Mahdy 2020, Limniou et al., 2021, Mahdy & Ewaida 2022, and Mahdy & Sayed reported greater use of the above-mentioned devices, while Saadeh et al., 2021 and Saadeh et al., 2022 revealed lower use when compared to the averages.

Figure 4 shows the textbook use among global veterinary students as assessed by only four studies of the mini-meta-analysis [19,21,41,42]. Similarly, e-book use was measured by only three studies conducted during the COVID-19 lockdown [19,21,22]. These results are shown in Figure 5. Four studies assessed the use of research papers (Figure 6) for study purposes among veterinary students [21,22,41,42].

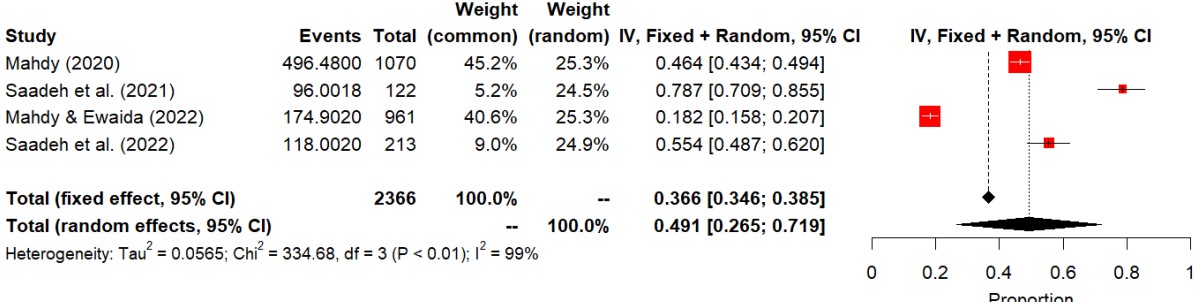

**Figure 4.** The proportion of veterinary students who use textbooks for learning purposes. Note: CI = confidence interval 95%; Tau$^2$ = index of effect size dispersion; Chi$^2$ = Cochran's Q (chi-squared) test of heterogeneity; df = degree of freedom; P = *p*-value; I$^2$ = Higgins statistic.

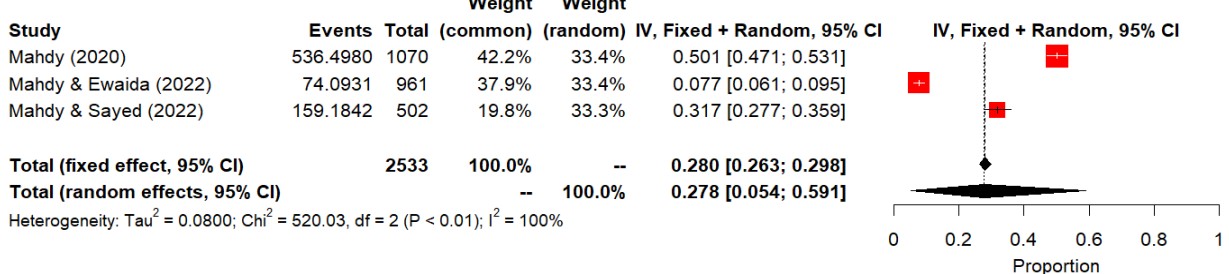

**Figure 5.** The proportion of veterinary students who use e-books for learning purposes. Note: CI = confidence interval 95%; Tau$^2$ = index of effect size dispersion; Chi$^2$ = Cochran's Q (chi-squared) test of heterogeneity; df = degree of freedom; P = *p*-value; I$^2$ = Higgins statistic.

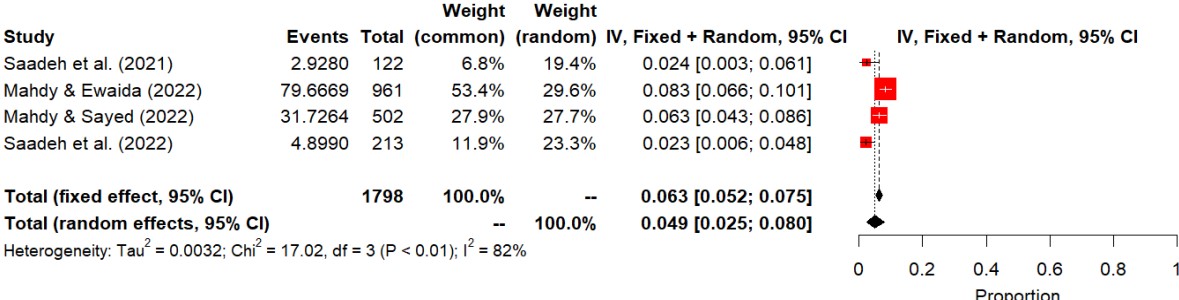

**Figure 6.** The proportion of veterinary students who use research papers for learning purposes. Note: CI = confidence interval 95%; Tau$^2$ = index of effect size dispersion; Chi$^2$ = Cochran's Q (chi-squared) test of heterogeneity; df = degree of freedom; P = *p*-value; I$^2$ = Higgins statistic.

Surprisingly, textbook usage results showed the maintained utilization of textbooks by veterinary students. The meta-analysis fixed effect model yielded an average value of 0.366 with a confidence interval of 0.346–0.385. On the other hand, the random effect model resulted in 0.491 [0.265–0.719]. Differences in results between the two models are due to the great sample size heterogeneity between analyzed studies. As shown by the figure, when compared to the average, Saadeh et al., 2021 and Saadeh et al., 2022 demonstrated higher use, while Mahdy 2020 and Mahdy & Ewaida 2022 reported lower use.

The proportion of veterinary students who used e-books for their learning yielded an average use of 0.280 with a confidence interval of 0.263–0.531 (fixed effect model), while the random effects model showed an overall value of 0.278 [0.54–0.591]. Compared to the average, Mahdy 2020 and Mahdy & Ewaida 2022 reported higher use of e-books among veterinary students, while Mahdy & Sayed 2022 reported lower use.

Finally, veterinary students use few research papers for their study; in fact, the fixed meta-analysis model resulted in 0.063 [0.052–0.075], while the random effects model resulted in 0.049 [0.025–0.080]. Compared to the average, Mahdy & Ewaida and Mahdy & Sayed exposed greater use of scientific papers by veterinary students, while Saadeh et al., 2021 and Saadeh et al., 2022 revealed minor use of scientific papers by veterinary students to support their learning.

Five studies evaluated the use of educational websites (Figure 7) among veterinary students [19,21,22,41,42]. The use of educational applications (apps) was assessed by only three studies [19,21,22], and the results are shown in Figure 8. Lastly, the use of YouTube videos and social media platforms was assessed by a total of five studies [19,21,22,41,42], and they are graphically presented by Figures 9 and 10, respectively.

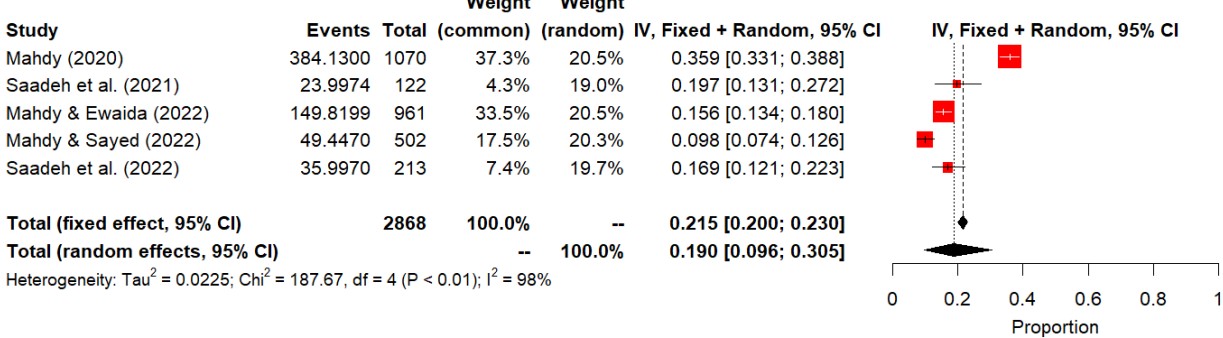

**Figure 7.** The proportion of veterinary students who use educational websites for learning purposes. Note: CI = confidence interval 95%; Tau$^2$ = index of effect size dispersion; Chi$^2$ = Cochran's Q (chi-squared) test of heterogeneity; df = degree of freedom; P = *p*-value; I$^2$ = Higgins statistic.

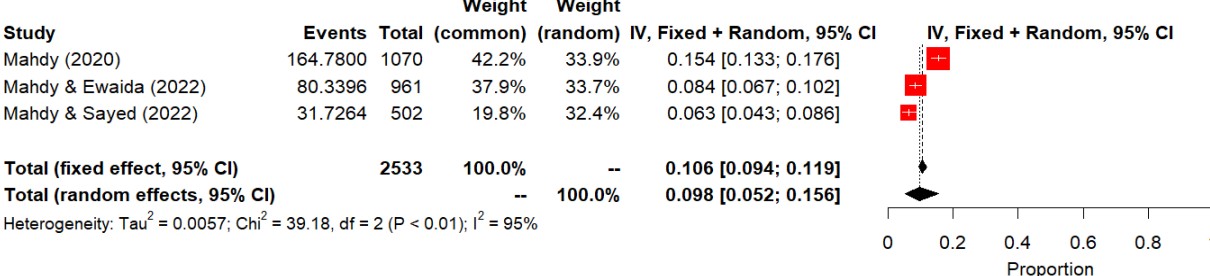

**Figure 8.** The proportion of veterinary students who use educational applications for learning purposes. Note: CI = confidence interval 95%; Tau$^2$ = index of effect size dispersion; Chi$^2$ = Cochran's Q (chi-squared) test of heterogeneity; df = degree of freedom; P = *p*-value; I$^2$ = Higgins statistic.

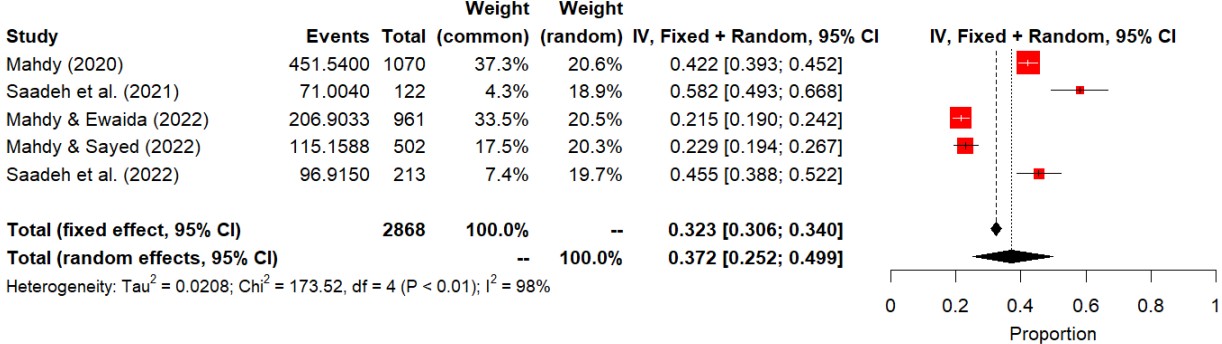

**Figure 9.** The proportion of veterinary students who use YouTube videos for learning purposes. Note: CI = confidence interval 95%; Tau$^2$ = index of effect size dispersion; Chi$^2$ = Cochran's Q (chi-squared) test of heterogeneity; df = degree of freedom; P = *p*-value; I$^2$ = Higgins statistic.

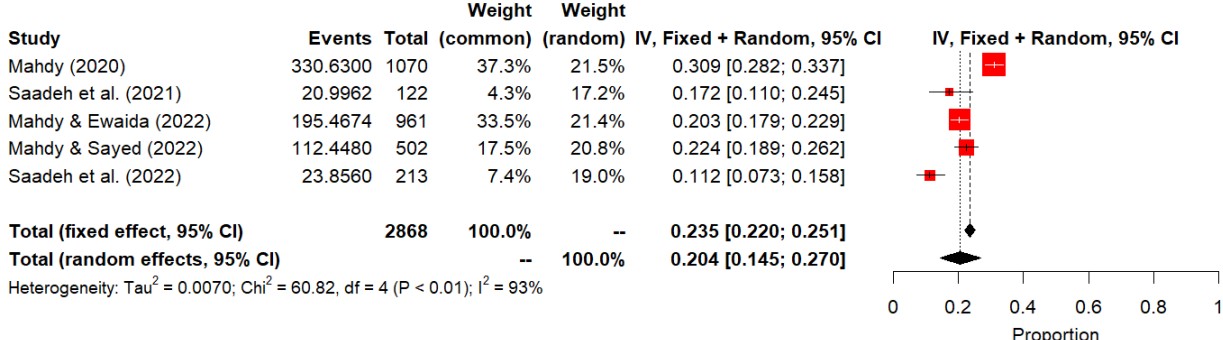

**Figure 10.** The proportion of veterinary students who use social media platforms for learning purposes. Note: CI = confidence interval 95%; Tau$^2$ = index of effect size dispersion; Chi$^2$ = Cochran's Q (chi-squared) test of heterogeneity; df = degree of freedom; P = *p*-value; I$^2$ = Higgins statistic.

The selected studies showed moderate use of all these online tools by veterinary students. However, considering the diversity provenience of the students enrolled in the analyzed studies, these results were expected.

Findings with regard to the meta-analysis fixed effects model of the use of educational websites among students reported an average value of 0.215 [0.200–0.230]; meanwhile, the random effects model reported 0.190 [0.096–0.305]. When compared to the average, only Mahdy 2020 reported greater use of educational learning websites. Saadeh et al., 2021, Mahdy & Ewaida 2022, Mady & Sayed 2022, and Saadeh et al., 2022 resulted in similar use.

Regarding the use of educational applications, an average use of 0.106 with a confidence interval [0.133–0.176] was reported by the fixed effect model and a total value of 0.0098 [0.052–0.156] was reported by the random effect model. In general, all the studies

reveal similar uses of educational applications. More specifically, Mahdy 2020 reported higher use of educational applications.

Meanwhile, for YouTube videos, the fixed effect model yielded an overall value of 0.322 [0.306–0.340]. The random effect model showed an overall value of 0.372 [0.252–0.499]. In general, all the studies reported a similar proportion of YouTube video use for educational purposes among students. More specifically, Mahdy 2020, Saadeh et al., 2021, and Saadeh et al., 2022 revealed higher use of YouTube videos when compared to the average. Conversely, the study of Mahdy & Ewaida 2022 and Mahdy & Sayed 2022 reported lower use of YouTube videos if compared to the average.

Finally, regarding the use of social media platforms, the fixed effect model resulted in an average use of 0.235 with confidence interval [0.220–0.251] and a total value of 0.204 [0.145–0.270] in the random effect model. In general, all the studies reported similar use of social media platforms for study purposes among students. Only Mahdy 2020 reported higher use of social media platforms when compared to the average. However, all the studies reported similar results for social media use.

## 4. Discussion

To the best of our knowledge, this is the first systematic review and mini-meta-analysis of the cross-sectional studies that have explored the use of online resources and electronic devices among global veterinary students. Our inclusion criteria limited the number of the selected papers and allowed us to identify only six studies eligible for the mini meta-analysis. This indicates that few studies are conducted in veterinary education, despite the relevance of this area. Our study selection reveals that the majority of studies were conducted in the last few years. It is plausible to think that the pandemic situation could have also influenced this field of research, but it also reveals the growing focus on veterinary higher education and interest in how new digital ways of teaching are rapidly impacting veterinary curricula. It should be noted that although the number of studies is small, these studies were all published in the last 5 years despite a 10-year analysis interval.

The present meta-analysis reports a significant use of electronic devices for learning purposes among veterinary students. On the one hand, the usage of non-portable media devices varied significantly across countries. More specifically, the present findings show that the population of veterinary students that reported the lowest proportion of non-portable electronic devices usage (approximately 2.4%) derived from Mahdy & Sayed 2022, which included 502 participants in the first or second year of 17 Egyptian veterinary schools [22]. In addition, lower usage of non-portable devices was also reported in two multi-national studies [19,21]. A possible explanation for these findings is that the economic status of veterinary students varies between countries. Since approximately more than 60% of respondent students [19,21] were from less-developed countries, it is to be expected that more affordable devices would have been used. Therefore, when designing electronic educational materials, veterinary educators should be aware of what devices their students are more likely to use. Saadeh et al., 2021 revealed a greater use of non-portable devices among United Kingdom students [41,42]. One possible reason explaining this finding may lie in the expandability and ease of repair in these devices compared to portable media devices.

On the other hand, 54.5% to 90.6% of students use portable electronic devices such as smartphones, laptops, and tablets for learning purposes. This widespread use of portable devices might be explained by the fact that they are easily accessible, allow quick access to information, and are less expensive than non-portable devices. Portable media devices enable the use of online tools during face-to-face classes, which can lead to better student engagement and more interactive classes.

In addition, free Wi-Fi connection allows students to reduce their costs. Due to these characteristics, portable media devices can facilitate opportunistic learning, which may be of great interest for commuter veterinary students in particular. According to the veterinary educational literature, the use of portable media devices in learning environments

has fostered collaboration between veterinary students [43] and enhanced their learning experience [44]. Furthermore, research also shows that usage of portable electronic devices in higher education courses increases student enjoyment, attention, and learning [45]. In addition, the use of mobile devices can facilitate students' self-directed learning due to the affordability, accessibility, portability, and educational benefits of these devices [46].

When considering the use of textbooks, different results were obtained among the selected studies. The lowest proportion of students using textbooks derived from the multinational studies of Mahdy 2020 and Mahdy & Ewaida 2022 [19,21]. However, those studies analyzed data from periods of COVID-19 restrictions when, due to the social distance measures to avoid the spread of the viral infection, students could not physically access the libraries, an ordinary place where they usually get textbooks for their learning. An aspect that should not be underestimated is that university textbooks are generally expensive. Economic impact plays a crucial role in choosing textbooks as a learning source because they represent a significant percentage of expenses faced by university students [47]. A recent study in veterinary education has indicated that student willingness and ability to purchase textbooks are significantly influenced by the textbook costs [48]. Therefore, in selecting learning resources, teachers should consider the growing opportunity for electronic textbooks and/or ensure the availability of a consistent number of textbooks at the campus/school/department library.

Despite the many advantages of e-books, including portability and cost effectiveness [49], their use rated as slight to moderate among veterinary students. A possible explanation for this finding could be the limited availability of e-books compared with the traditional ones. Although the availability of digital learning materials is growing in the higher education market, the concept of the traditional printed textbook remains steadfast. The student preference for printed textbooks over e-books is confirmed by a large body of literature in higher education [50,51]. Therefore, veterinary educators should pay attention to student preferences when recommending different formats of books within veterinary educational settings.

Among the resources analyzed, the current meta-analysis results report low use of the research papers by veterinary students for their learning. These findings may be partially explained by the fact that students probably become more familiar with research papers in the last years of their studies or post-graduation [52], mainly when they prepare their final dissertation and they are actively engaged in research. However, it should be noted that it is probably necessary to support the use of scientific articles from the early years of the course, with a focus on the problem-solving approach that is now considered a more effective active method of learning. Thus, veterinary students should rapidly obtain the necessary skills to locate and identify relevant research papers [53] in order to acquire critical problem-analysis skills. A recent multi-dimensional survey study has reported that veterinary students encounter various difficulties in reading scientific papers. These include low confidence in their appraisal skills, difficulty in understanding statistics, or insufficient instruction in interpreting scientific papers [54]. Hence, due to the importance of revised information that research papers generally offer, the evidence-based veterinary medicine approach should be encouraged and teachers should identify strategies (e.g., journal clubs) to teach students the necessary skills to review and interpret research paper results.

The use of educational learning websites varies among studies according to different social contexts. In detail, the proportion of students who have reported to use educational websites for their learning purposes has ranged from 9.85% [22] up to 19.67% [41]. Conversely, there is limited use of educational websites by veterinary students, and this could be partially explained by the fact that these resources are relatively new within veterinary education. However, as is being reported in the medical education literature, the development of educational websites is likely to grow over time with advancements in technology, aiming to incorporate the requirements and the needs of today's students [34,55].

The use rate of educational apps for learning ranged from 6.3% among Egyptian students [22] up to 15.4% among multinational students coming from 92 different countries [19].

Considering that the study sample sizes were approximately 502 and 1392 respondents, it seems reasonable to assume that educational apps are little-used among veterinary students. One possible explanation for this finding could be that some apps are not free and may require payments and subscriptions. Another possible reason could be that some educational apps may have limited relationship to learning outcomes. Outside veterinary curricula, several educational apps have been developed to supplement student learning in interactive ways [56]. Therefore, veterinary establishments should include the purchase of educational apps in their budget, stimulate the creation of specific apps [57], and/or collaborate with other institutions to implement shared applications, as recent research has demonstrated that they are effective tools in increasing student knowledge and clinical skills [58,59].

The use of information from short video tutorials available on social media platforms, in particular on YouTube, is assuming considerable importance. According to Roshier et al. (2011), veterinary students had a positive perception of video usage, which increases in particular before practical examinations [60]. In addition, Müller et al. (2019) found that instructional videos are a valuable learning tool in veterinary education as they help students learn clinical skills and they contribute to animal welfare [18]. Veterinary educators and curriculum planners should promote the use of educational videos as a powerful resource to learners and as a means to reduce the number of animals used for educational activities [18,60]. Furthermore, videos allow unlimited access to learning material, and hence, veterinary learners can revisit them as often as they desire and can use them anytime, anyplace [61].

However, considering the number of students involved in these different studies, it is important to note that despite the widespread use of social media for leisure among university students, their use for educational purposes is limited. These findings are consistent with a previous systematic review and meta-analysis performed in medical education, which found that only 20% of medical students used social media sites for educational purposes [62]. However, studies within veterinary education have demonstrated that social media platforms are essential tools for collaborating with peers and sharing knowledge between them [3,41,42,63,64]. Social media is an excellent platform for sharing information, ideas, experiences, and videos, and it can provide students with creative learning methods [65], facilitate collaborative learning and student engagement, and increase student academic performance [66,67].

## 5. Limitations

It is necessary to acknowledge the limitations of the present study. First, the inclusion criteria limited the number of the selected papers and allowed us to identify only six studies eligible for the mini-meta-analysis. This study was limited to peer-reviewed English-language publications, and the grey literature was excluded. Moreover, most of the studies were too heterogeneous. There are possible reasons for the heterogeneity, such as the difference in the overall quality of veterinary students in different countries across the world. Another limitation is that this mini-meta-analysis investigated the usage of digital devices and learning resources and could not move beyond into, for example, ascertaining the efficacy of such use. Therefore, future studies should investigate the efficacy of the above-mentioned devices and learning resources in terms of outcomes among veterinary students.

## 6. Conclusions

The authors provide a mini-meta-analysis of recent publications evaluating the use of digital media devices and learning resources by veterinary students. This topic is of great interest since the evolution of technology, coupled with the advent of pandemic-related restrictions on in-person lessons, has made it imperative that educators consider how students may access educational material, as well as what type of educational materials may be available to them.

Our results suggest that even though we live in a technologically advanced world, veterinary student attitudes towards digital resources cannot be assumed. Digital readiness is needed, both from a student and a teacher perspective. Veterinary establishments should increase their efforts to purchase and/or develop digital education tools that are specifically targeted at facilitating knowledge transfer among veterinary students.

**Author Contributions:** Conceptualization, E.M., D.C. and E.V.; methodology, E.M. and D.C.; writing—original draft preparation, E.M., D.C. and E.V.; writing—review and editing, R.O., M.B. and D.B.; supervision, E.M., D.C. and E.V. All authors have read and agreed to the published version of the manuscript.

**Funding:** This research received no external funding.

**Institutional Review Board Statement:** Not applicable.

**Informed Consent Statement:** Not applicable.

**Data Availability Statement:** Not applicable.

**Acknowledgments:** Smartsheep, Fondazione Cassa di Risparmio di Cuneo (CRC), Agriculture 4.0.

**Conflicts of Interest:** The authors declare no conflict of interest.

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
