# Peer review of "Are Veterinary Students Using Technologies and Online Learning Resources for Didactic Training? A Mini-Meta Analysis"

_education, doi:10.3390/educsci12080573_

Round 1

Reviewer 1 Report

A well written and thorough review/meta-analysis of the use of technology in veterinary education. The study design is sound and the information is very timely and relevant for veterinary educators. 

Author Response

Dear Reviewer,

I would like to thank you for very much your precious time and effort as well four your consideration. Its a honour reading your positive comments towards our paper.

Thank you very much!

Kind regards,

Edlira

Reviewer 2 Report

The aim of the current study is to conduct a systematic review and a mini meta-analysis of the cross-sectional studies that have explored the use of online resources and electronic devices among global veterinary students.  Results highlight a high use of portable media devices with differences among countries, still good use of traditional textbooks, moderate use of online tools and low use of research papers. The conclusions suggest that Digital readiness is needed both from a student and a teacher perspective, therefore Veterinary Establishments should increase their efforts to purchase and/or develop digital education tools specifically targeted to facilitate the knowledge transfer in veterinary students. 

In general, this manuscript and methodology chosen for the research are suitable for publication.

The research topic is meaningful and relevant, as few studies have been conducted in veterinary education.

The manuscript is clear and presented in a well-structured manner; the figures/tables/ schemes are appropriate and easy to interpret and understand.

The theoretical framework presents bibliographical references relevant and up-to-date, although it is suggested that specific ones on the topics of online learning and higher education be integrated (particularly in relation to the topic of university teaching in the Covid-19 pandemic).

The theme (use of electronic devices for learning purposes among veterinary students) is enough consistent with the journal scope and the conclusions are interesting for the readership (however limited number of people, i.e., of the specific context of veterinary education). 

Therefore, it is suggested that:

1) integrate some literature references into the theoretical framework related to distance learning in higher education, so as to attract a wide readership and anchor concluding reflections (currently too general) to studies on online teaching/learning in higher education (a much more relevant topic)

2) integrate limitations and prospect of the study: although the discussion of the data is relevant, the conclusions could be strengthened by indicating less generically some perspectives (curricular, methodological, educational) for strengthening the digital competence of teachers and students in veterinary courses; It would also be useful to highlight the limitations and future prospects of the study.

Specific comments referring to line numbers:

·      line 53: delete an endpoint (there are two after "students")

·      line 92, first line of Table 1: check double and/or triple parentheses

·      line 195/201: check the overlap section 3.2 Study Characteristics with Figure 1

Thanks for this interesting reading.

Author Response

Dear Reviewer,

I would like to thank you very much for your precious comments and your time and effort. Please see the attachment in the present box.

Thank you very much for helping us to improve the quality of our work.

Best regards,

Edlira
